# Functions of coastal feeder bluff systems: Implications for prioritizing protection and restoration

Jason D. Toft[1]*, Julia N. Kobelt[1], Kerry L. Accola[1], Megan N. Dethier[2], Andrea S. Ogston[3], Sarah E. Vollero[3]

1 School of Aquatic and Fishery Sciences, University of Washington, Seattle, Washington, United States of America, 2 Friday Harbor Laboratories, University of Washington, Friday Harbor, Washington, United States of America, 3 School of Oceanography, University of Washington, Seattle, Washington, United States of America

* tofty@uw.edu

## Abstract

Actively eroding cliffs, known as feeder bluffs, are important sources of sediment for coastal beaches. When shorelines have artificial armor, natural beach sediment processes can be disrupted. A recent management tool in the Salish Sea, WA, USA prioritizes efforts with high potential benefit of restoration (armor removal) or protection (preventing armor construction) to nearshore sediment supply. We conducted field sampling at 20 beaches identified as the highest or lowest priorities for restoration or protection, so we could examine local ecological and physical functions, in addition to potential landscape benefits. We sampled parameters spanning the top of the bluff to the low shore, and evaluated a total of 30 metrics including riparian vegetation, invertebrate assemblages, logs, beach wrack, fish abundance and behavior, surface epifauna and algae, beach and bluff characteristics, and sediment size and sorting. For analyses, we calculated an average score of beach function for each of four treatments: "Protect High" (unarmored, ranked as high management priority), "Protect Low" (unarmored, ranked low), "Restore High" (armored, ranked high), and "Restore Low" (armored, ranked low). Protect High and Low treatments were equivalent in local beach function, and both scored over twice as high as Restore treatments. Restore High scored only slightly higher than Restore Low, indicating a consistent degradation caused by armoring on bluffs with variable sediment source potential. Statistical models revealed that overall beach function may be largely driven by upper beach metrics including wrack, logs, and overhanging vegetation. Metrics for geomorphology and lower beach organisms were more variable, likely due to differences in geographical region and distance from the bluff. Our results indicate that beaches with natural unarmored bluffs have the highest level of localized ecological function regardless of the level of potential sediment supply, and restoring sediment supply processes at beaches with armored bluffs could double their ecological function.

**Data availability statement:** All data files are available from the Dryad database: Dataset DOI: 10.5061/dryad.vx0k6dk3j Reviewer URL: http://datadryad.org/share/FkrvWaUtI_dRwxO1hy-iUmr3gHQpszuY4aE0YRyrPs_w.

**Funding:** Funding was provided by an ESRP learning project (PRISM #20-1932). https://wdfw.wa.gov/species-habitats/habitat-recovery/puget-sound/esrp The funders had no role in study design, data collection and analysis, decision to publish, or preparation of the manuscript.

**Competing interests:** The authors have declared that no competing interests exist.

## Introduction

Coastal cliffs occur along about 80% of the world's shorelines [1], and the connections to the adjacent beach are critical to the physical and ecological functioning of the nearshore zone. Feeder bluffs are defined as erosional high-elevation coastal cliffs that supply (i.e., feed) sediment to the nearshore [2,3]. These systems have been emphasized as priority conservation habitats worldwide [4], and are therefore of management concern for protection and restoration. For example, in Britain there are concerns about rising sea levels leading to detrimental impacts to the extent of the intertidal zone at the base of cliffs [5], with associated risks of coastal erosion and inundation of neighboring low-lying land [6,7]. Conservation efforts have focused on revegetation with native plants and reintroduction of bluff associated fauna, e.g., the endangered El Segundo Blue Butterfly in California, USA [8]. In addition to physical and ecological functions, coastal bluffs can also provide aesthetic and cultural value [9].

Erosional feeder bluffs in the Salish Sea, WA, USA are identified as targets for both protection and restoration [10]; their diverse conditions regionally enable evaluation of the ecological and physical functions of the connected bluff and beach systems. Puget Sound comprises the southern extent of the Salish Sea, and its morphology was shaped by the repeated advance and retreat of glaciers during the Pleistocene [3]. Today, the bluffs that line most of the approximately 4000 km of Puget Sound shoreline [10] are composed of poorly sorted but well consolidated glacial till, more easily eroded glacial marine drift, and varying pre-Vashon age fluvial units [11]. Bluffs are considered to be the primary sediment source of the littoral (or drift) cells, which are relatively self-contained reaches along the coast that contain a complete sedimentation cycle: source, paths of transport, and sinks [12]. Transport and sorting of sediment by currents and waves occur along the beach below the bluff and downdrift [2,13]. Ultimately, sediment sinks can be located in the nearshore, e.g., low elevation accretion shoreforms where sediments are deposited, or can be carried offshore into deeper basins.

Beyond "feeding" sediment to downdrift habitats, the ecological functions of feeder bluffs for surrounding biota can be multifaceted. Direct positive associations include pigeon guillemots (*Cepphus columba*) nesting on the bluff face and feeding on fish in the surrounding area [14], and forage fish spawning on intertidal beach sediments [15]. The beaches below erosional feeder bluffs have been shown to have a higher proportion of surface sand and number of fallen trees than shorelines associated with non-erosional shoreforms [16]. Other perceived functions have important food web implications yet have less direct data documentation. For example, shallow water areas are used as nursery habitat by juvenile Chinook salmon (*Oncorhynchus tshawytscha*) that are listed as threatened under the federal Endangered Species Act, by chum and pink salmon (*O. keta* and *O. gorbuscha*), and also by forage fish such as Pacific herring (*Clupea pallasii*) [17]. Juvenile Chinook salmon also feed on insects along the shore [18], thus connecting the marine-terrestrial interface.

In Puget Sound, 29% of shorelines are armored (i.e., seawall, bulkhead, rip-rap) in order to stabilize banks and protect human infrastructure [19], similar to elsewhere

along the U.S. coastline [20] and worldwide [21,22]. Armor has been shown to have negative regional [23] and global [24] impacts on shoreline ecology, leading to efforts to provide guidance on restoration priorities for removing armor. This is no small task, given the projected global increase in marine built structures [25]. The best documented impacts of shoreline armor in the Salish Sea are in upper intertidal-supratidal areas [23], which are also where the most effective restoration (e.g., armor removal) occurs for wrack, logs, sediment, vegetation, and invertebrates [16,26–28]. For coastal landforms such as feeder bluffs, armor removal often restores these features to natural levels, but restoration response can be dependent on the underlying shoretype and wave climate [16].

Within the Puget Sound waters of the Salish Sea, the 'Beach Strategies' project by the Washington Department of Fish and Wildlife's Estuary and Salmon Restoration Program (ESRP) provides a suite of science-based tools to guide the prioritization of restoration and conservation opportunities [10,19]. One of the foci is on sediment supply processes at the drift cell (landscape) scale [10]. Bluffs categorized as "restore" had artificial armor at the base of the bluff, while "protect" bluffs had no human modifications. In the ESRP framework, a site with a low amount of degradation of sediment supply due to shoreline armoring, and a high potential benefit from restoration, is recommended as "restore high", i.e., a good candidate for leveraging restoration resources. A site that has a heavily degraded sediment supply with a lower potential benefit from restoration is recommended as "restore low". Similarly, sites with no armor and high existing sediment supply benefits are designated as "protect high", and those with lesser existing benefits are designated as "protect low". The premise of this prioritization is that the top-ranked bluffs (to either restore or protect) will have a stronger influence in providing sediment to down-drift beaches, and in turn may also rank high on a scale bar of localized beach functions.

Our project builds on this decision support tool by connecting geomorphology to ecological function at bluffs prioritized for restoration and protection. By focusing on the function of bluffs and connections to the adjacent beaches, our work facilitates building a more holistic approach to the evaluation framework and thus increases the probability of optimal outcomes. Improved knowledge of these bluff-beach systems should be applicable to many similar systems worldwide [1,2] and to their associated management guidelines. Our two main objectives are to (1) assess whether bluff sites prioritized for sediment supply also have the highest local realized ecological functions, beyond potential downdrift sediment benefits; and (2) determine what integrated geomorphic factors govern ecological functions of bluff-beach coastal systems, and how armor may impact these functions.

## Materials and methods

### Sampling approach and study area

Study sites were selected from the bluffs prioritized for restoration and protection by ESRP [10], through an evaluation of metric scores across Puget Sound. We conducted field sampling at twenty sites, five each ranked as (1) Restore High, (2) Restore Low, (3) Protect High, and (4) Protect Low (S1 Table). Our twenty sites were dispersed equally among five geographic regions in Puget Sound (Fig 1), with each region encompassing sites of all four treatments. We focused our site selection on calculated numerical potential benefits to sediment supply processes [10]. In order to incorporate a broad range of values, and have more sites for selection within a given region, we included some that were labeled as "moderate" when potential benefit scores were near low rankings (S1 Table). Potential sediment supply benefit was higher and process degradation lower at the highly ranked sites, and vice versa for the lower ranked sites (S1 Table). We used ESRI ArcGIS (version 10.8.2) to query site options within each region, and verified site suitability and access prior to fieldwork. We hereafter refer to a site or bluff as a general term; the four "Treatments" of Protect High (PH), Protect Low (PL), Restore High (RH), and Restore Low (RL) (Fig 2); and the five "Regions" of Penn Cove, Dabob Bay, Agate Pass, Vashon Island, and Harstine Island.

### Metrics and data collection

For each of our 30 surveyed ecological and physical metrics, we assigned a rationale based on the goals of our project such that a higher value indicates greater beach function (S2 Table, values were inverted in cases where lower values

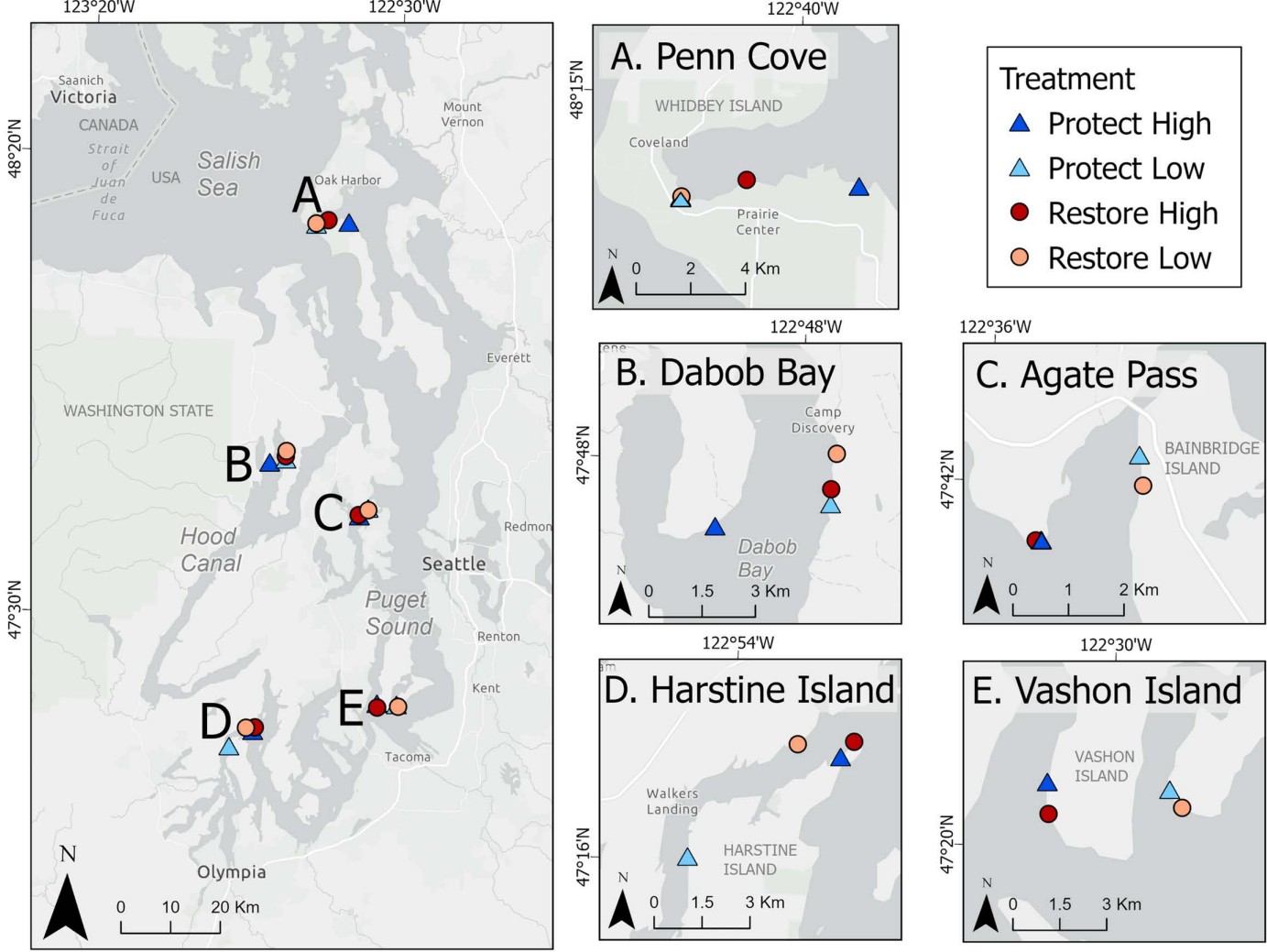

**Fig 1. Map of field collection sites in Puget Sound and Hood Canal, the southern reaches of the Salish Sea.** The four treatments are detailed with circle and triangle symbols, and the five regions are labeled on the map with separate panels: (A) Penn Cove; (B) Dabob Bay; (C) Agate Pass; (D) Harstine Island; (E) Vashon Island. Also see photos of treatments in Fig 2, and specific site locations detailed in S1 Table. Map created in ArcGIS Pro 3.5, basemap sources: Esri, TomTom, Garmin, FAO, NOAA, USGS, (c) OpenStreetMap contributors, and the GIS User Community. Map image is the intellectual property of Esri and is used herein under license. Copyright © 2025 Esri and its licensors. All rights reserved.

indicate greater beach function). We surveyed metrics during spring-summer 2022, from the bluff to the low shore, using established protocols, many documented further in the Shoreline Monitoring Database [29]. Sampling was generally conducted along 50 m transects parallel to shore or as space allowed, as some sites were limited by length of armor or natural bluff. We scaled random sampling along each transect to be comparable across sites.

We measured percent cover of wrack deposited on the beach during the ebbing tide by using a 0.1 m² quadrat at ten random points along each transect. We apportioned the wrack into algae, eelgrass (*Zostera* spp.), and terrestrial components. At each quadrat location we also measured beach wrack width (m) and depth (cm).

We characterized logs on the beach that were sourced locally (fallen trees) and regionally (driftwood deposited after ebbing tides and/or storms). At five random points along each transect, we surveyed the width of the log line (m), counted

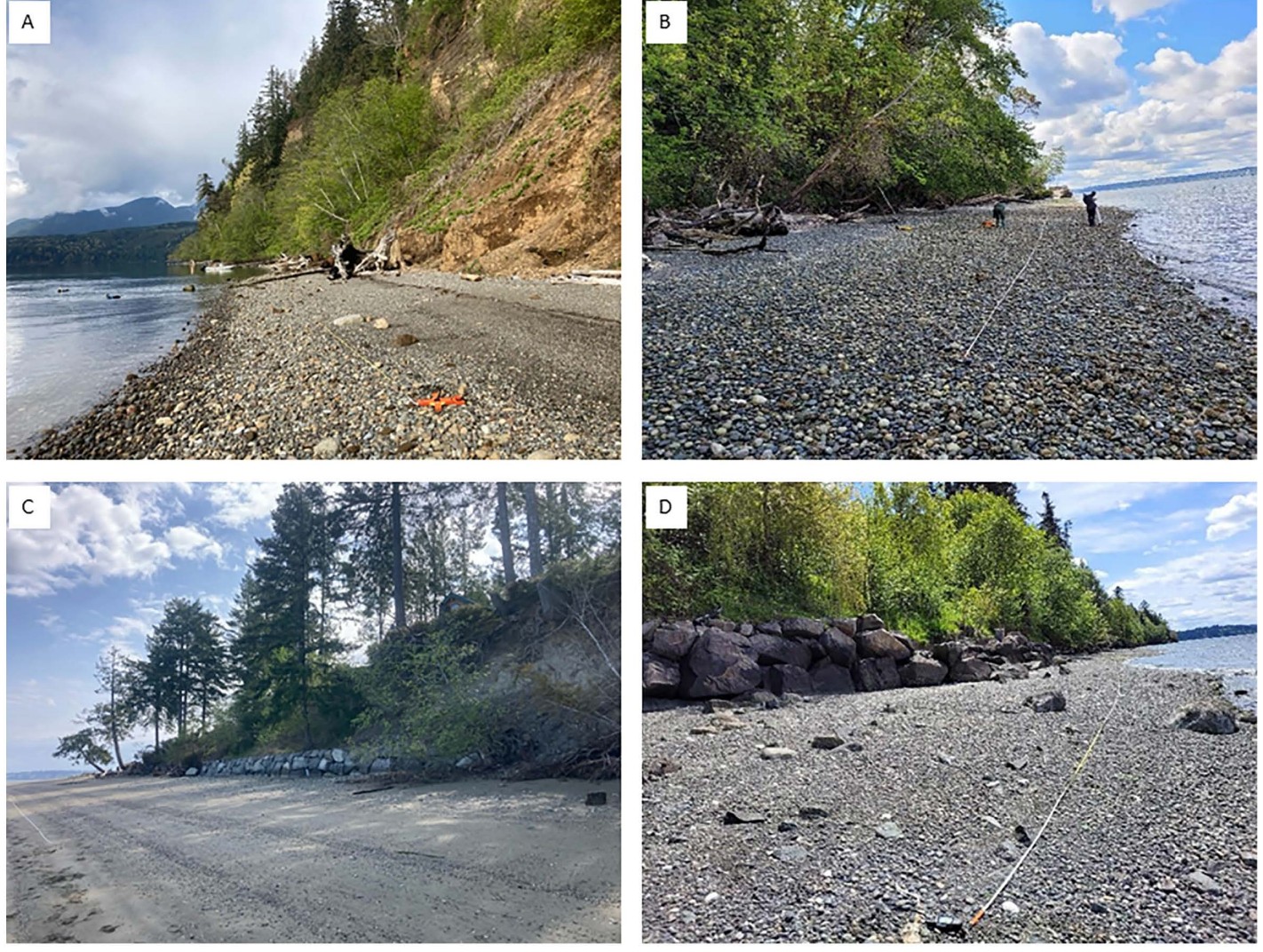

**Fig 2. Fieldwork site photos.** (A) "Protect High" natural bluff in Dabob Bay; (B) "Protect Low" natural bluff on Vashon Island; (C) "Restore High" armored bluff on Harstine Island; (D) "Restore Low" armored bluff in Agate Pass.

the number of logs (longer than 1 m and above 10 cm diameter), and noted if they were partially buried by sediment. We also counted the total number of fallen trees.

We used snorkel surveys to measure fish and crab densities and feeding behavior, surveying at high tides to ensure that fish were close to feeder bluffs and shoreline features. We surveyed each site twice, spring and summer, to span the peak outmigration period of juvenile salmon. During each sampling event, four transects at depths spanning intertidal areas were surveyed by snorkeling parallel to shore for up to 75 m, or as space allowed. Counts were converted to densities by incorporating transect length and underwater horizontal visibility as measured by a Secchi disk. Underwater videos were used for verification of fish identifications and behavior.

Along each transect, we measured percent cover of overhanging vegetation (e.g., red alder *Alnus rubra*) and supratidal vegetation (e.g., dunegrass *Elymus mollis*). Overhanging vegetation was calculated by measuring start and end linear distances of trees overhanging the beach. When dunegrass was present, we measured up to five subplots for patch width

(m). Plants were identified to develop a species list of vegetation located within 5 m of the backshore, and species placed into native and non-native categories.

At seven random points along each transect, we used fallout traps to sample densities and composition of supratidal invertebrates, i.e., insects and other arthropods. Fallout traps were $40 \times 25$ cm plastic bins with a small amount of soapy water to relieve surface tension, left out for two hours. Samples were preserved in isopropanol and returned to the laboratory for microscope processing of taxa and number.

We gathered data on abundances of flora and fauna visible at the surface or under surface cobbles (e.g., algae, seagrass, barnacles, snails, and crabs) and sediment grain sizes at Mean Lower Low Water (MLLW) using ten 0.25 m$^2$ quadrats along a 50 m transect and standard SCALE methodology [30]. These surface parameters are indicative of the more complex community (e.g., clams and worms) found in the underlying sediment.

Drone imagery, flown at ~15.2 m above the beach foreshore, was collected to create cm-scale orthophotos and digital elevation models of beach and bluff morphology, as well as to estimate the height and other characteristics of bluffs. Imagery was visually evaluated to estimate the area of exposed sediment within the bluff face located behind the transect (bluff exposure). A real-time kinematic global positioning system (RTK-GPS) was used to collect elevation profiles of beaches perpendicular to shore, from the base (toe) of feeder bluff or armor to below MLLW. The RTK-GPS was also used to record ground control points (GCP) for image processing, and to locate Mean Higher High Water (MHHW), Mean Sea Level (MSL), and MLLW sampling areas at each site. We focused on the beach slope from bluff toe to MSL for analyses, as it coincided with upper beach metrics and proximity to bluff or armor. We calculated the relative encroachment of armor or bluff into the local tidal frame using the elevation of the toe relative to the MHHW tidal elevation. Specifically, the relative encroachment is defined as the difference between MHHW and the toe elevation of the bluff or armor for the site; positive encroachment indicates the toe elevation is below that of MHHW, encroaching into intertidal areas, whereas negative encroachment indicates the toe elevation is above MHHW, allowing for continued beach gradient into supratidal areas. We analyzed detailed photographs of sediment in quadrats (0.64 m$^2$) placed at the surface and 5 cm subsurface to visually characterize the percent area of different grain sizes, and used Gradistat [31] for statistical analysis at two elevations: MHHW and MLLW.

We evaluated wave characteristics and potential for alongshore transport of sediment using observed climate data [32], and an evaluation of fetch along the dominant wind pathways. Winds were characterized by data from the Seattle-Tacoma International Airport (for the main basin of Puget Sound), and the Olympia Airport (for the South Sound). For each treatment site within the five regions, the open-water fetch distance was mapped along the dominant wind direction applicable to that site (i.e., between 169 and 214° North for southerly facing sites; and 349 and 34° North for northerly facing sites in the main basin). The angle between the dominant fetch vector and the shoreline was determined. Storm wave height was evaluated assuming a 6-hr duration wind event of 20 km/hr if the site faced S/SW, 15 km/hr if N/NE, and 10 km/hr if in a direction of infrequent winds. An estimated significant storm wave height and period were calculated from these values, using a simplified method for predicting wave characteristics in deep water based on [33] as presented in [34].

## Data analyses

We first created an overall scale bar of beach functions by (1) averaging treatments across regions for each measured metric; (2) scaling the averages from 0 to 1 by the min and max values within each metric; and (3) averaging the scaled values across metrics, producing an overall value of beach function at each of the four treatments.

Subsequently, we analyzed response variables that had replication within each treatment using generalized linear mixed models (GLMM) (e.g., ecological measurements such as beach wrack, as detailed in data collection methods above), and those with one data point at each treatment with analysis of variance (ANOVA) (e.g., physical measurements such as beach slope). We performed post-hoc pairwise comparisons of estimated marginal means from GLMMs and Tukey honest significant difference tests of ANOVA. Variables included beach wrack (percentage total cover, percentage

terrestrial cover, wrack depth and line width), logs (total counts, width of log line, partially buried logs, fallen trees), fish (total counts, juvenile salmon counts and feeding behaviors, forage fish densities), vegetation (percentage overhanging, percentage supratidal, dunegrass patch width, native vegetation species richness), supratidal invertebrates (fallout trap taxa richness and densities), MLLW taxa richness and percentage eelgrass, wave height, geomorphology (slope and width, bluff height, relative encroachment, and exposure), and sediments (percentage sand and sorting at MHHW and MLLW).

All analyses included treatment as a factor, and potential covariates included relative encroachment of the bluff, beach slope and width, and variables appropriate to each analysis (e.g., percent overhanging vegetation for terrestrial wrack, percent sand for MLLW biota, and water depth for fish analyses). Covariates were selected based on the questions being addressed and by using corrected Akaike Information Criterion [35] to examine best model fit. Site location in Puget Sound (i.e., region) was incorporated as a random effect to account for variation among regions [36], and fish models incorporated random effects of season (spring/summer). Transect lengths (or area) were scaled for fish counts (or densities). We used a Chi-squared test to analyze if there was a difference among treatments in the frequency of instances when juvenile salmon were observed feeding versus not feeding. We also used multivariate tests on assemblages of supratidal invertebrates and MLLW biota.

Percentage responses were transformed into proportion responses modeled by the beta family. Continuous responses followed a Gaussian distribution, while count data followed a Poisson or negative binomial distribution. GLMM, ANOVA, and Chi-square statistical analyses were performed using R Statistical Software [37]. Data manipulation was performed using "dplyr" [38]. We built GLMMs using R package "glmmTMB" [39], pairwise comparisons using the package "emmeans" [40] and used "ggplot2" [41], sjPlot [42], and kableExtra [43] for data visualization. Multivariate analyses were conducted using the Primer and PERMANOVA programs [44] with a Bray-Curtis resemblance matrix. Data were log-transformed for supratidal invertebrates and square root transformed for MLLW biota before multivariate analysis, with taxa representing less than 3% of the total abundance of any one sample removed [45]. Structure of multivariate analyses were conducted as for GLMM analyses above, along with vectors of correlation on NMDS plots.

## Results

### Metric overview and scale bar

We collected data on 30 metrics that informed our overall scale bar of local beach function (Table 1). When metrics were scaled from 0 to 1 and averaged, beach function was two to three times greater at the natural Protect High and Low treatments than the armored Restore High and Low (respectively) (Fig 3).

### Wrack

Overall, wrack metrics were higher at Protect treatments than Restore treatments (Figs 4A-D, Tables 1, 2, S3 Table). Both the Protect High and Protect Low treatments had significantly higher total percent wrack cover than Restore High, while there were no differences with Restore Low (Fig 4A). However, for the terrestrial component of the wrack, Protect High and Low were significantly higher than both Restore High and Low (Fig 4B). Protect High had significantly higher wrack depths than the other treatments, and both Protect High and Low had higher wrack depths than Restore High and Low (Fig 4C). Protect Low had a significantly wider wrack line than the other treatments, and Restore High was wider than Restore Low (Fig 4D).

Relative encroachment improved model fits for wrack cover (S3 Fig, Table 2), significantly so for wrack width (Fig 5A, Table 2); wrack lines were narrower with armor or bluff toe encroachment below MHHW. Slope of the upper beach significantly improved model fits for wrack cover and width (Figs 5B and 5C, Table 2); wrack cover decreased as slope steepened, while wrack width increased as slope steepened.

**Table 1. Average values for each metric of treatment across regions.**

| Parameter | Metric | Protect High | Protect Low | Restore High | Restore Low | *Treatment differences* |
|---|---|---|---|---|---|---|
| *Wrack* | % Wrack Cover | 43.2 | 41.7 | 13.7 | 24.0 | PH & PL>RH |
| | % Terrestrial Wrack | 10.1 | 11.3 | 1.5 | 2.7 | PH & PL>RH & RL |
| | Wrack Depth (cm) | 2.6 | 2.0 | 1.1 | 1.1 | PH>PL>RH & RL |
| | Wrack Width (m) | 4.0 | 4.9 | 3.4 | 2.4 | PL>PH & RH & RL; RH>RL |
| *Logs* | Log Count | 2.6 | 3.4 | 0.1 | 0.0 | PH & PL>RH & RL |
| | Width of Log Line (m) | 1.9 | 3.3 | 0.1 | 0.0 | PH & PL>RH & RL |
| | Count Partially Buried Logs | 0.8 | 1.4 | 0.1 | 0.0 | PH & PL>RH & RL |
| | Fallen Tree Count | 13.0 | 8.8 | 1.2 | 0.0 | PH>PL>RH & RL |
| *Fish* | Total Fish Density (100m$^{-2}$) | 52.6 | 2.6 | 34.6 | 8.7 | PH>PL & RH & RL; RH>PL |
| | Juvenile Salmon Density (100m$^{-2}$) | 14.8 | 0.4 | 6.2 | 3.1 | PH>PL & RH & RL |
| | Forage Fish Density (100m$^{-2}$) | 91.2 | 0.0 | 55.5 | 5.6 | *No differences* |
| | Juv. Salmon Feeding Observations (%) | 5.9 | 0.0 | 28.6 | 50.0 | *Overall difference* |
| *Vegetation* | % Overhanging Vegetation | 89.8 | 97.0 | 42.2 | 27.6 | PH & PL>RH & RL |
| | % Supratidal Vegetation | 15.0 | 34.0 | 3.2 | 0.0 | *No differences* |
| | Dunegrass Patch Width (m) | 0.1 | 1.9 | 1.1 | 0.0 | PL>PH & RL |
| | Native Vegetation Taxa Richness | 12.6 | 14.0 | 8.0 | 6.8 | *No differences* |
| *Supratidal* | Fallout Trap Taxa Richness | 3.4 | 3.9 | 3.8 | 2.9 | *No differences* |
| *invertebrates* | Fallout Trap Density (m$^{-2}$) | 57.7 | 67.4 | 73.1 | 136.3 | RL>PH |
| *MLLW biota* | Biota taxa richness | 6.5 | 5.8 | 6.2 | 5.2 | *No differences* |
| | % Eelgrass | 2.0 | 1.2 | 1.0 | 19.1 | PH & RL>PL & RH |
| *Geomorphology* | Beach Slope (m/m) Toe-MSL | 0.11 | 0.12 | 0.11 | 0.14 | *No differences* |
| | Beach Width (m) Toe-MLLW | 41.8 | 49.7 | 39.5 | 39.0 | *No differences* |
| | Bluff Height (m) | 28 | 10 | 15 | 19 | *No differences* |
| | Bluff Exposure approx. (m$^2$) available | 1685 | 370 | 45 | 0 | PH>RH & RL |
| | Relative Encroachment (m) | −0.51 | −0.12 | 0.27 | 0.41 | *No differences* |
| | Wave Height (m) | 0.48 | 0.48 | 0.45 | 0.48 | *No differences* |
| *Sediment* | % Sand MHHW | 36 | 42 | 59 | 44 | *No differences* |
| | % Sand MLLW | 60 | 64 | 58 | 60 | *No differences* |
| | Normalized Sorting MHHW (unitless) | 5.4 | 6.5 | 2.3 | 3.8 | PL>RH |
| | Normalized Sorting MLLW (unitless) | 3.1 | 3.3 | 3.7 | 2.2 | *No differences* |

Shading represents a scaling of high to low, where darker colors are qualitatively higher on the scale. Treatment differences are a summary of quantitative post-hoc pairwise statistical tests, including Tukey tests of ANOVAs and pairwise comparisons of estimated marginal means from GLMMs. Treatment abbreviations are Protect High (PH), Protect Low (PL), Restore High (RH), and Restore Low (RL). If a specific treatment is not listed, then there were no pairwise significant differences with that treatment.

## Logs

Similar to wrack, log metrics were overall higher at Protect treatments than Restore treatments (Figs 4E-H, Tables 1, 2, S3 Table). Restore Low treatments had all values of zero for log count, width of the log line, partially buried logs, and fallen trees. Protect High and Low had significantly higher values than Restore High for all those same log metrics. Protect High additionally had more fallen trees than Protect Low.

Relative encroachment improved model fits for partially buried logs, significantly so for log count and width of the log line (S3 Fig, Table 2). However, this response was almost all due to the Protect treatments due to the paucity of values besides zero at the Restore treatments, and therefore did not add much beyond the Treatment effects, as toe elevation of bluffs at Protect treatments averaged above MHHW (see geomorphology results below) and model visualizations showed

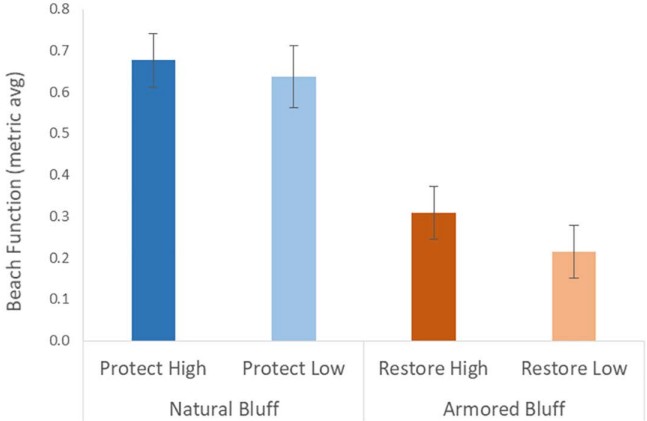

**Fig 3. A scale bar of our measured beach function at the four treatments.** "Protect" are natural bluffs, "restore" are armored bluffs, with high and low rankings based on potential benefits to sediment supply processes [10]. Values are averages of scaled metrics from 0 to 1 across our 30 measured metrics, with higher values indicating greater beach function. Error bars are Standard Error.

high variability centered around zero. Slope of the upper beach improved model fits for log line width, although not significantly (S3 Fig, Table 2), and showed that log width decreased as slope steepened.

## Fish

Fish densities were generally higher at Protect High and Restore High treatments, and lower at the Low treatments (Figs 6A and 6B, S1 Fig, Tables 1, 2, S3 Table). Protect High had significantly higher total fish counts than the other treatments, and Restore High was higher than Protect Low. The majority of fish observed on snorkel surveys were surfperches (Embiotocidae), forage fish (Pacific sand lance *Ammodytes hexapterus*, northern anchovy *Engraulis mordax*, and Pacific herring), and juvenile salmonids (chum, pink, Chinook, coho *Oncorhynchus kisutch*, and trout *Oncorhynchus* spp.), with a mix of other fish and crabs (S4 Fig). Juvenile salmon and forage fish accounted for most of the taxa observed in spring, while surfperches and forage fish were most frequently observed in summer (S4 Fig).

Protect High had significantly higher juvenile salmon counts than the other treatments (Fig 6B, Tables 1, 2, S3 Table). Average minimum lengths of juvenile salmon ranged from 2.5 cm for chum and pink, up to 12.5 cm for Chinook, coho, and trout (S4 Table). Juvenile salmon were mostly located in the surface to middle of the water column, more so than other fish (except for forage fish) (S4 Table). Juvenile Salmon were observed to feed predominantly in surface and middle waters, except for juvenile chum salmon, which also fed along bottom substrates (S5 Table). Juvenile salmon feeding behavior was most often observed at Restore Low (6 out of 12 observations) and Restore High (4 out of 10 observations), rarely at Protect High (1 out of 16 observations) and was not observed at Protect Low (0 out of 7 observations) (overall Chi-squared $p = 0.016$). There were no significant treatment differences in forage fish counts (S1 Fig, Table 1, S3 Table), although they were observed at all treatments except Protect Low.

## Vegetation

Percent of overhanging vegetation was significantly higher at Protect High and Low than Restore High and Low (Fig 6C, Table 1, S3 Table). The top five species of overhanging vegetation by total linear extent were red alder (32%), Pacific madrone (*Arbutus menziesii*) (16%), Douglas fir (*Pseudotsuga menziesii*) (13%), bigleaf maple (*Acer macrophyllum*) (9%), and willow (*Salix* spp.) (7%) (S6 Table). Percent supratidal vegetation had a trend towards being higher at Protect treatments, and was zero at Restore Low treatments, but there were no statistically significant differences (S1 Fig, Table 1, S3 Table). Dunegrass

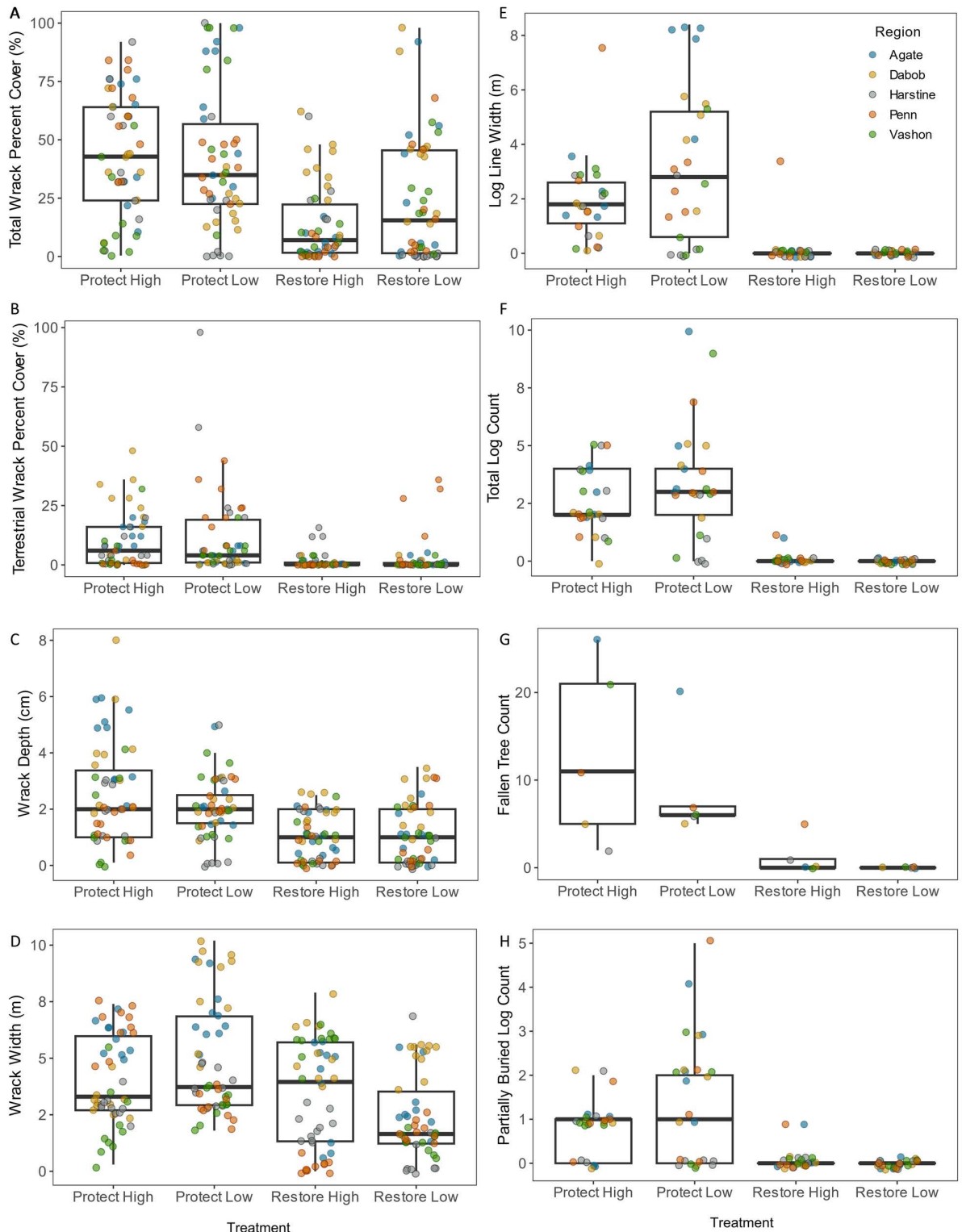

**Fig 4. Boxplots of wrack and logs metrics with significant Treatment differences.** (A) Wrack percent cover; (B) Terrestrial wrack percent cover; (C) Wrack depth; (D) Wrack width; (E) Log line width; (F) Total log count; (G) Fallen tree count; (H) Partially buried log count. Data points are color-coded by Region, and are minimally staggered to avoid overlap.

**Table 2. Generalized linear mixed model summaries for metrics with sufficient data.**

| | Predictors | (Intercept) Protect High | Protect Low | Restore High | Restore Low | Relative Encroachment | Slope (Toe-MSL) |
|---|---|---|---|---|---|---|---|
| **Wrack Proportion** | Estimates | 1.76 | 0.97 | 0.34 | 0.61 | 0.75 | 0.89 |
| | SE | 0.75 | 0.21 | 0.09 | 0.16 | 0.12 | 0.04 |
| | p | 0.189 | 0.888 | **<0.001** | 0.065 | 0.072 | **0.003** |
| **Terrestrial wrack Proportion** | Estimates | 0.09 | 1.04 | 0.44 | 0.43 | | |
| | SE | 0.01 | 0.2 | 0.09 | 0.08 | | |
| | p | **<0.001** | 0.817 | **<0.001** | **<0.001** | | |
| **Wrack Depth** | Estimates | 2.63 | −0.61 | −1.57 | −1.56 | | |
| | SE | 0.28 | 0.22 | 0.22 | 0.22 | | |
| | p | **<0.001** | **0.006** | **<0.001** | **<0.001** | | |
| **Wrack Line Width** | Estimates | 2.23 | 1.24 | −0.02 | −1.11 | −0.72 | 0.14 |
| | SE | 0.92 | 0.37 | 0.43 | 0.48 | 0.33 | 0.07 |
| | p | **0.016** | **0.001** | 0.958 | **0.021** | **0.027** | **0.04** |
| **Total Logs** | IR Ratios | 3.13 | 1.09 | 0.02 | 0 | 1.49 | |
| | SE | 0.47 | 0.21 | 0.02 | 0 | 0.25 | |
| | p | **<0.001** | 0.636 | **<0.001** | 0.998 | **0.02** | |
| **Log Line Width** | Estimates | 4.23 | 1.27 | −2.55 | −2.92 | 1.11 | −0.17 |
| | SE | 1.08 | 0.58 | 0.89 | 1.12 | 0.53 | 0.1 |
| | p | **<0.001** | **0.029** | **0.004** | **0.009** | **0.035** | 0.093 |
| **Partially Buried Logs** | IR Ratios | 1.08 | 1.32 | 0.06 | 0 | 1.64 | |
| | SE | 0.27 | 0.41 | 0.05 | 0 | 0.43 | |
| | p | 0.772 | 0.379 | **<0.001** | 1 | 0.063 | |
| **Total Fish** | IR Ratios | 130.22 | 0.12 | 0.30 | 0.37 | | |
| | SE | 36.74 | 0.05 | 0.12 | 0.16 | | |
| | p | **<0.001** | **<0.001** | **0.002** | **0.024** | | |
| **Total Juvenile Salmon** | IR Ratios | 60.66 | 0.07 | 0.23 | 0.1 | | |
| | SE | 77.23 | 0.05 | 0.11 | 0.06 | | |
| | p | **0.001** | **<0.001** | **0.003** | **<0.001** | | |
| **Fallout Trap Taxa Richness** | IR Ratios | 3.29 | 1.18 | 1.15 | 0.84 | | |
| | SE | 1.18 | 0.15 | 0.15 | 0.12 | | |
| | p | **<0.001** | 0.187 | 0.282 | 0.219 | | |
| **Fallout Trap Densities** | IR Ratios | 53.41 | 1.35 | 1.3 | 2.4 | | |
| | SE | 11.54 | 0.35 | 0.32 | 0.64 | | |
| | p | **<0.001** | 0.253 | 0.291 | **0.001** | | |
| | | | | | | % Sand | |
| **Taxa Richness MLLW** | IR Ratios | 9.28 | 0.95 | 0.95 | 0.87 | 0.99 | |
| | SE | 1.27 | 0.08 | 0.08 | 0.07 | 0 | |
| | p | **<0.001** | 0.495 | 0.514 | 0.092 | **<0.001** | |
| **Proportion Sand MLLW** | Estimates | 1.43 | 1.02 | 1.04 | 1.1 | | |
| | SE | 0.37 | 0.24 | 0.26 | 0.26 | | |
| | p | 0.169 | 0.923 | 0.867 | 0.688 | | |

Values in the table include estimates or incidence rate (IR) ratios, standard errors, and p-values (highlighted in bold when significant p < 0.05). Models compare means of multi-level categorical variables (e.g., the difference between "Protect Low" and the intercept "Protect High") and calculate the effect of continuous predictors on the response. IR ratios are the logged quotient of the difference of expected counts, where each level of categorical variable is compared to the intercept ("Protect High"). An IR ratio of 1 indicates group incidence rates are equal. Blank cells indicate the predictor is not included in the models. All models include random effects of region. Fish models incorporate random effects of season (spring/summer) and area, and MLLW metrics include % sand.

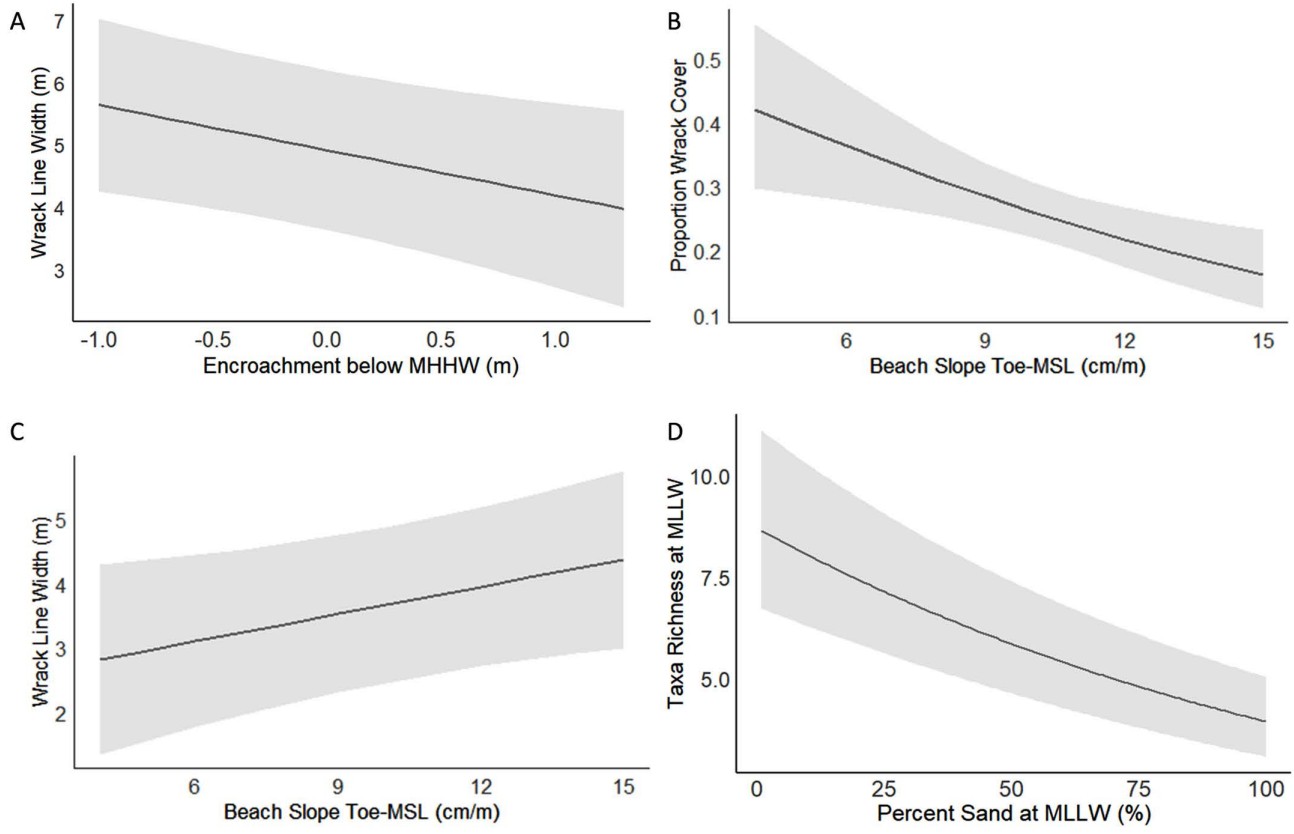

**Fig 5. Line plots of significant predictors in mixed effects models.** (A) Wrack line width by encroachment below MHHW; (B) Proportion wrack cover by beach slope; (C) Wrack line width by beach slope; (D) Taxa richness of biota by percent sand at MLLW. 95% confidence intervals are shaded gray.

patch width was significantly higher at Protect Low than Protect High and Restore Low, with no differences at Restore High, and was zero at Restore Low treatments (Fig 6D, Table 1, S3 Table). Much of the dunegrass recorded at Restore High treatments was growing within or above the armor, not in beach sediments. Native vegetation taxa richness had a trend towards being higher at Protect treatments, but there were no statistically significant differences (S1 Fig, Table 1, S3 Table). The most frequently recorded native species were red alder, honeysuckle (*Lonicera* spp.), oceanspray (*Holodiscus discolor*), Pacific madrone, and dunegrass, and the most frequent introduced species were Himalayan blackberry (*Rubus armeniacus*), sowthistle (*Sonchus* spp.), English ivy (*Hedera helix*), and scotch broom (*Cytisus scoparius*) (S5 Fig).

## Supratidal invertebrates

Taxa richness in the fallout traps had a trend towards being lowest at Restore Low, but there were no significant differences, although the pairwise difference between Protect Low and Restore Low was marginally significant ($p = 0.055$) (S1 Fig, Tables 1, 2, S3 Table). Overall densities in the fallout traps were significantly higher at Restore Low than Protect High (Fig 6E, Tables 1, 2, S3 Table). The majority of the taxa composition at Restore Low (over 80%) were non-flying arthropods – acari (mites) and collembola (springtails) – while the other treatments had more insects such as diptera (flies), hemiptera (true bugs), and hymenoptera (wasps, bees, and ants), which comprise primary juvenile Chinook salmon insect prey items (S6 Fig). PERMANOVA showed treatment was significant ($p < 0.0001$), with all pairwise comparisons significantly different from each other ($p < 0.05$) except Protect High and Low.

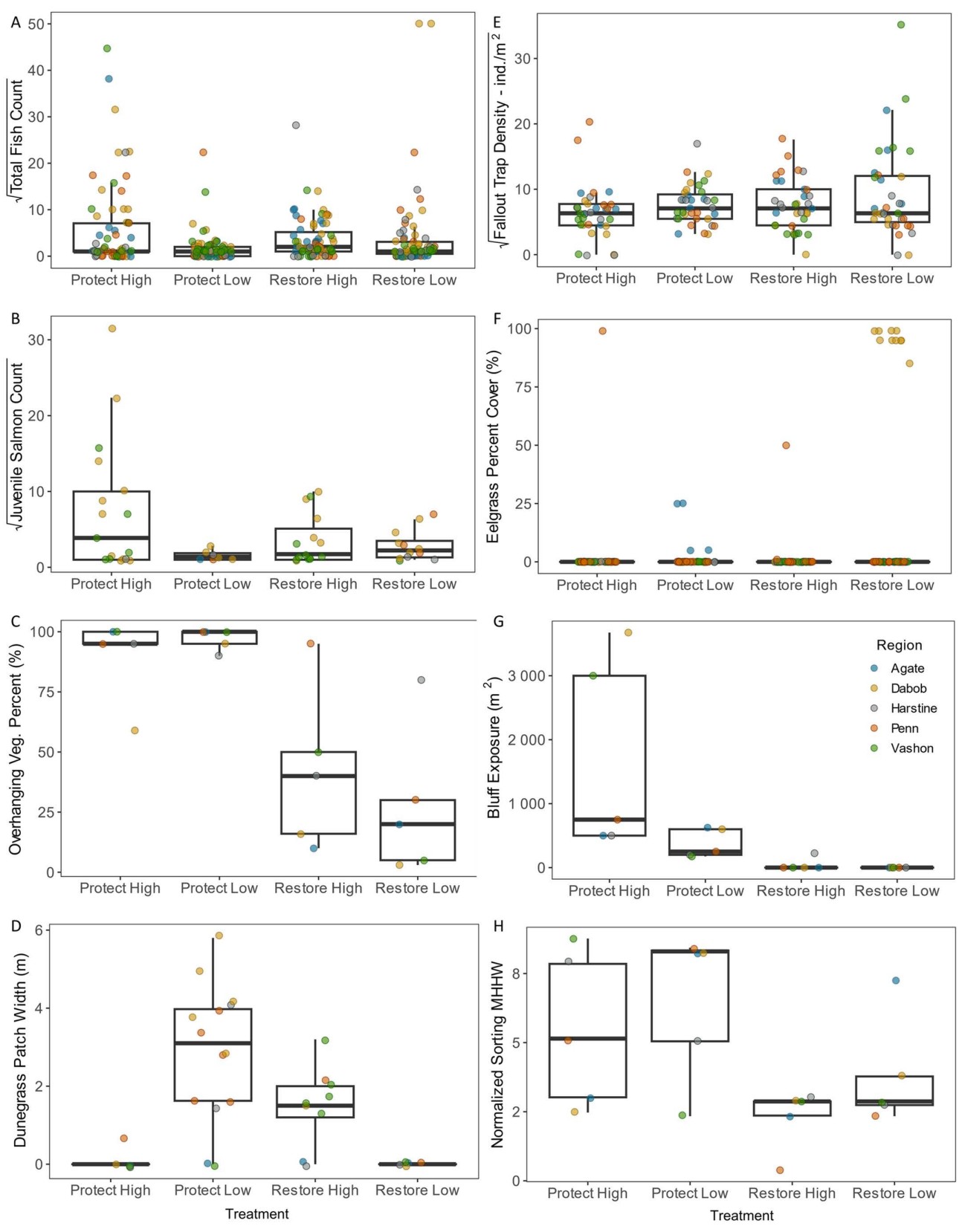

**Fig 6. Boxplots of fish, supratidal invertebrates, eelgrass, geomorphology, and vegetation metrics with significant Treatment differences.** (A) Total fish count; (B) Juvenile salmon count; (C) Bluff exposure; (D) Normalized sediment sorting at MHHW; (E) Dunegrass patch width; (F) Overhanging vegetation percent; (G) Fallout trap density; (H) Eelgrass percent cover. Data points are color-coded by Region, and are minimally staggered to avoid overlap.

## MLLW biota

Taxa richness of biota at MLLW had a trend towards being lowest at Restore Low, but there were no significant differences (S1 Fig, Tables 1, 2, S3 Table). Eelgrass was present at only 4 of the 20 sites, one of each treatment type across three of the regions. When eelgrass was present, percent eelgrass at MLLW was significantly higher at Protect High and Restore Low than Protect Low and Restore High, and was most abundant at the Restore Low treatment in Dabob Bay (Fig 6F, Tables 1, 2, S3 Table). Most of the eelgrass, including all of the eelgrass at the Restore Low treatment in Dabob Bay, was the introduced *Zostera japonica*. The native *Zostera marina* was recorded at the Protect High and Restore High treatments in Penn Cove. Taxa composition overall varied by Region, with Dabob Bay having more eelgrass and oysters than other regions (S7 Fig). Taxa composition also varied with sediment type, as sand was inversely related to cobble, pebble, oyster, and shell hash (S7 Fig), and an increase in percent sand decreased taxa richness (Fig 5D).

## Geomorphology

The upper beach slope from toe of bluff or armor to MSL tended to be lower at Protect High and Restore High, although differences were not significant (S2 Fig, Table 1, S3 Table). By geographic region, overall beach slopes from toe-MLLW were steepest at Harstine, least steep at Dabob Bay, and highly variable at the Vashon and Penn Cove regions between treatments (S8 Fig). Beach widths from toe of bluff or armor to MLLW were overall widest at the Protect treatments, specifically Protect Low, although differences were not significant (S2 Fig, Table 1, S3 Table).

Bluffs backing the beaches were up to 50 m in height. The tallest were found at the Protect High treatments, although differences were not significant (S2 Fig, Table 1, S3 Table). The approximate unvegetated bluff exposure was significantly greater at Protect High treatments than the Restore treatments, with no exposure at the Restore Low treatments, indicating that the Restore Low bluffs were completely vegetated (with limited potential for supply of material to the beach) (Fig 6G, Table 1, S3 Table).

Relative encroachment of armor tended to be below MHHW elevation for the Restore treatments (positive values), although differences were not significant (S1 Fig, Table 1, S3 Table). Even in regions where the Protect treatments had the toe of bluff below MHHW, they were always above the Restore treatments. Similarly, in regions where Restore treatment relative encroachment values were above MHHW (negative values), the Protect treatment values were more positive. Calculated storm wave heights and alongshore transport capacity showed no significant differences (S1 Fig, Table 1, S3 Table). Treatments within regions often had similar fetch and wind characteristics and thus, similar predicted wave heights (S8 Fig).

## Sediment

The mean grain size of the sediment on the studied beach sites was variable, ranging from silt ($D_{50} = 0.016$ mm) to coarse cobbles ($D_{50} > 250$ mm) (S2 Fig). We found no significant difference between treatments for percent sand at MLLW or MHHW (S2 Fig, Tables 1, 2, S3 Table). Grain size sorting, as characterized by a normalized sorting value, varied by elevation. There were no significant differences between the treatments at MLLW, which were poorly sorted (S2 Fig, Table 1, S3 Table). However, at MHHW we found that there was poorer sorting at Protect Low and Protect High treatments, significantly poorer at Protect Low than at Restore High (Fig 6H, Table 1, S3 Table).

## Discussion

Overall, we found that bluffs categorized as either Protect High or Low (natural with no armor) for shoreline management priority actions [10] had an average metric of beach function that was over twice as high as bluffs categorized as Restore High or Low

(with artificial armor). This scaling of functions provides information specific to feeder bluff/beach systems, building on previous work across a broader set of shoretypes in Puget Sound (i.e., including low elevation beaches [23]), and applicable to coastal cliffs elsewhere in the world [1,2]. Potential benefits of sediment supply to drift cells ranked as high and low within Protect and Restore treatments did not correspond to an equivalent range in beach functions – Protect High and Low were equal in their overall ranking of beach functions, and Restore High was only slightly higher than Restore Low. This signifies that unarmored bluffs have similar, high levels of local beach ecological function, regardless of the potential contribution of sediment from the surrounding drift cell or levels of shoreline degradation. Conversely, armored bluffs have more limited ecological functions that can be slightly influenced by local levels of sediment supply and degradation. There was, however, a range in the response of individual beach function metrics and variability across regions, especially in the category of geomorphic metrics.

### Geomorphology across regions and feeder bluff processes

While the focus of our project was to compare the four treatments, local geomorphology is often a function of the regional geological setting, leading to variability in wave energies and sediment sources. The sediment sources differ as bluffs within different regions of Puget Sound are made up of unique geologic units, and these vary in lithology, grain size distributions, and erodibility [3]. More broadly, bluffs in many coastal systems can be composed of stratigraphic sequences resulting in failure planes that impact the rate of bluff recession and angle of the exposed bluff face [1,2]. Different geologic sequences could therefore alter the need for armoring of the toe of the bluff for protection of human infrastructure. Although we do not address these factors, the glacial sediment exposed at our studied bluffs contained only limited evidence of failure planes due to organic or clay-rich sediment and generally had similar face exposures.

At the tidally fluctuating shoreline where the interaction between marine waters and the bluff toe occurs, wind-forced waves provide the primary contribution to erosive bed stress in Puget Sound [3]. For example, beaches located in protected embayments experience limited wave energies, which impacts the sediment characteristics at the bluff toe relative to more exposed environments [2]. In locations where the wave energy is greater, bluffs can more easily erode via notching and removal of colluvium at the toe. Treatments within a region varied in their placement within specific drift cells, thus creating variability in the sediment source contribution to alongshore transport. In addition, winds and waves can affect other habitat forming processes, e.g., a recent study in Puget Sound showed that sites with a larger fetch had higher input of deposited wrack and logs, whereas sites with a smaller fetch had higher input from localized terrestrial sources – fallen trees and eroding sand [16].

We expected to find that beaches backed by taller bluffs with exposed sediment on their surfaces would be broader and flatter in slope, with finer and more poorly sorted sediment, as is generally found in sandier beach environments [46]. Additional factors that can influence the rate of bluff failure and erosion are sediment composition and stratigraphic layering that control pore water pressure, and bluff incisions leading to weakness during heavy precipitation and frost or ice heave. Although these factors can influence the grain size and slope relationship on our studied mixed sediment beaches, effects were not clearly evident. However, at the armored (Restore) sites beach width tends to be narrower, and encroachment relative to MHHW is always more positive than at unarmored sites. The bluffs are more vegetated on the face (although vegetation does not overhang the shoreline), and therefore these bluffs have less available sediment to supply the beach. Sorting of the grain size distribution also appears to be affected at the Restore beaches, but only clearly at MHHW as those sediment distributions are more directly linked to the source materials in the bluff, and wave action is amplified near the armoring at the higher tidal elevations.

Armoring has been found to impact beach slope and grain size due to its effects on wave energy and erosion [47,48]. However, we found no significant differences in beach slope or mean grain size between the four treatments. Differences in slope and grain size related to increased wave energy at armored sites may not be observable due to regional variability. Elevation of armor placement also may govern the magnitude of response, as our Restore treatments had armor only in the upper portion of the tidal frame, with mean toe of armor 0.34 m below MHHW (range 1.26 m below to +0.8 m above). If armor had extended deeper into the tidal frame, responses would likely be heightened. Other aspects of armor

that may affect the observed impact include time since placement and type of material [23]. These should be considered in future regional mapping and management of sediment supply and related ecological function.

In open coast studies, mean grain size and beach slope have been found to be related [49–51]. However, a recent summary analysis of 2144 beaches [52] found that for coarse-grained beaches (grain size larger than ~2mm), slope and grain size are not strongly related. Beach slopes and mean grain size in our study fall within the range of data evaluated in [52], and our data are consistent with the finding that there is not a strong relationship between beach slope and grain size on these coarser grained beaches.

## Feeder bluff functions

Our ecological measurements of the upper beach predominantly had higher values at the unarmored Protect treatments than the armored Restore treatments, albeit with a few exceptions. Measurements of beach wrack, deposited logs and fallen trees, and overhanging vegetation were always higher at Protect treatments. These are key habitat characteristics of a healthy upper beach system. In shoreline riparian areas where there is overhanging vegetation, temperature and light levels are moderated, improving success of forage fish beach spawning [15], and partially buried logs are a stabilizing feature for sediments and plant colonization [53]. Supratidal invertebrates had high variability across all sites; taxa richness along the shoreline edge had a trend of lower at Restore low treatments, while overall densities were higher at Restore Low sites than Protect High. Taxa composition is relevant to the interpretation of this finding, as at Restore Low, the majority of invertebrates in the fallout traps were not insects, but non-flying arthropods – acari and collembola – while the Protect treatments had more insects such as diptera, hemiptera, and hymenoptera, which are key juvenile Chinook salmon prey items [18,54].

The inclusion of additional parameters beyond treatment in model selection improved our understanding of upper beach functions, sometimes in non-intuitive ways. Given the overall low values of wrack and especially log measurements along armored Restore treatments, we may only be able to interpret signatures of relative encroachment to MHHW and upper beach slope in regard to Protect treatments (natural bluffs). At steeper upper beach slopes, wrack cover was lower, although the overall wrack-line width increased down the beach face. Conversely, along shallower slopes there was more wrack cover, but a more narrowly defined wrack line. Additionally, encroachment below MHHW of armor or bluff decreased the wrack line width. This may signify that overall sources of wrack are similar across treatments, but the spatial distribution of the wrack deposited on the beach is affected by the beach morphology, as well as being impacted by artificial armor [55]. Log counts and width of the log line at Protect treatments increased as the toe of the bluff approached MHHW. However, the average toe elevation was above MHHW, signifying that logs can settle along natural bluffs when ample upper beach habitat is available, as opposed to where armor placement almost completely prevents accumulation of logs.

Moving offshore, our fish observations showed a mixed response across Protect and Restore treatments, variability that likely encapsulates how a mobile fish reacts directly to aquatic habitat features (e.g., water depth, eelgrass), versus indirectly to features of the terrestrial landscape. Protect High treatments had the highest densities of total fish and juvenile salmon, although there was no difference between Protect Low and the Restore treatments, and forage fish showed no difference across treatments. We saw the most incidents of feeding behavior at Restore Low treatments, indicating that juvenile salmon are feeding along armored shorelines when they can inhabit, even with reduced densities, the adjacent nearshore habitat. This feeding response may be related to the high elevation placement of armor at our study sites, which allowed for most of the intertidal area to be inundated at high tides with minimal truncation by armor placement. Fish inhabiting these shallower water depths close to shore point to the importance of shallow water as nursery habitat [56]. By furthering our understanding of beach functions for juvenile fish, we can better incorporate the ecological role of feeder bluffs into species recovery plans through protection and restoration. Fish species that were a focus of our surveys included juvenile Chinook salmon that are listed under the federal Endangered Species Act, and Pacific herring that are a candidate for species of concern in Washington State; both are important targets for restoring and protecting shoreline habitats.

Our study design included sampling of biota and sediments low on the shore (at MLLW) to examine patterns associated with feeder bluff processes. While we found clear ties between sediment type (e.g., cobble vs. sand), dominant species (e.g., ulvoid algae vs. eelgrass), and region (e.g., oysters were dominant in Dabob Bay but not elsewhere), the variance in low-shore biota appeared to relate minimally to the characteristics of the feeder bluff or the degree of degradation of sediment supply. For example, taxa richness of biota was similar across treatments, and eelgrass abundance was variable, being most prevalent at only one site. Furthermore, most of the eelgrass was the introduced *Zostera japonica*; the native *Zostera marina* was only recorded at Protect High and Restore High treatments in one of the regions. This may lend credence to the high ranking to restore or protect those sites, in order to preserve native species, and the beneficial habitat functions that beds of *Z. marina* can provide [57,58]. As with our geomorphology measurements, variation across regions can be pronounced and may limit our ability to detect consistent signals across large spatial scales. For example, in a more limited regional study, taxa richness of lower intertidal biota was found to be higher at feeder bluffs than at accretional beaches and modified shorelines [59].

These results in the low shore are similar to those found in broader studies of shoreline features, where strong responses to human modifications were found on the upper shore but not the lower [16,23]. Sediment characteristics on the low shore, which is usually at some distance from the bluff, appear to be affected more by local wave energy and regional bluff composition, and potentially by the degree of sediment movement from unknown distances updrift. Sediment and wave energy, in turn, control the character of the biota; in general, sand-dominated low shore zones are species-poor except when eelgrass is present to bind the sediments and provide three-dimensional structure for other species, whereas cobble beaches provide habitat and sediment stability that benefit many benthic species. Thus, while sand abundance is considered a "positive" function in our study and is clearly positive at MLLW in terms of enabling growth of valuable eelgrass with a host of habitat benefits [57], it can have a "negative" function, e.g., on infaunal diversity. Interpretation thus depends on the habitat features and restoration or protection objectives of the site or project being quantified.

## Protection and restoration implications

Our focus on protection and restoration opportunities at feeder bluffs builds on previous efforts investigating the impacts of armor [23] and measurements of restoration effectiveness [16,26,27]. Strategies aimed at protecting natural bluffs can expect that an equivalent level of local ecological function will exist across high and low rankings of potential sediment supply benefit. The long-term equilibrium between bluff-sourced sediment and beach function, independent of the magnitude of the source, appears to be the important factor to protect. Strategies aimed at removing armor at the base of bluffs, in cases when the protection of upland human infrastructure is not substantiated or can be alleviated, can anticipate a local restoration trajectory that approaches the function of natural bluffs, thus potentially doubling the pre-restoration ecological function. We encourage more thorough documentation of functions through time given the dynamic nature of these bluff-beach systems, especially as related to any measurements of future restoration effectiveness and the limitations of our one year field study. Although Restore Low bluffs are ranked as a lower priority in the ESRP Beach Strategies tool for their potential contribution to sediment supply, they may generate the most gain for local ecological function, given their overall lowest ranking in our collected beach function metrics and therefore the greatest potential for amount of habitat rehabilitation. By evaluating both potential landscape scale sediment supply benefits along with local realized ecological function, managers can weigh the benefits of each in regard to their protection and restoration goals. These issues are not unique to the glacially carved Puget Sound with its coarse-grained beaches. For example, a program in Chesapeake Bay, USA, has used a similar framework of coastal conditions to provide management recommendations on where restoration actions are suitable [60], and protection of cliffs has been emphasized as a priority ecosystem worldwide due to the high biodiversity of species in cliff habitats [4].

Given our findings, we can offer some concrete examples of how restoration actions may incorporate additional features along feeder bluffs besides armor removal (along with related actions such as sediment nourishment and managed retreat), often a goal along Puget Sound shorelines [61] and elsewhere [62]. Although dunegrass in supratidal areas was highest at Protect Low treatments, there were occurrences of dunegrass along armored Restore High treatments, where much of the

dunegrass was in and above the seawall or riprap armor. This may present opportunities in a living shoreline context, where dunegrass may be able to be planted along shoreline armor in locations where the armor must be maintained for infrastructure needs. When incorporating logs into restoration, we can aim for natural conditions by partially burying logs for beach stability and plant growth, rather than placing logs on the surface of a beach. Fallen trees and overhanging vegetation can shade the shoreline and provide habitat structure – a beneficial feature of not only steep bluffs, but low-lying coastal forests as well, with similar issues at the seaward-landward boundary with vegetation stability and sea-level rise [63]. Although juvenile salmon densities were highest at Protect High treatments, we have shown that juvenile salmon can feed at high rates along armored shorelines when the armor is placed high on the shore, which supports current management guidelines of minimizing impacts of armor by placing it high above the shore and not into intertidal areas [64].

By linking data on beach functions to bluff management actions of restoration or protection, we will be able to assess future project performance on a data-driven scientific foundation. As armored bluffs are targeted for funding of restoration actions, our project provides baseline conditions to help measure effectiveness of those actions – e.g., when a seawall is removed at the base of a bluff, we can assess if a broader sediment distribution and more logs are deposited, the beach slope evolves, and juvenile salmon use is increased. Our analyses also evaluate the range of conditions at bluffs proposed for protection (e.g., source material and sediment types, and linkages to ecological functions). A conceptual model (Fig 7) illustrates linkages of field measurements and analyses to localized beach functions. By improving our

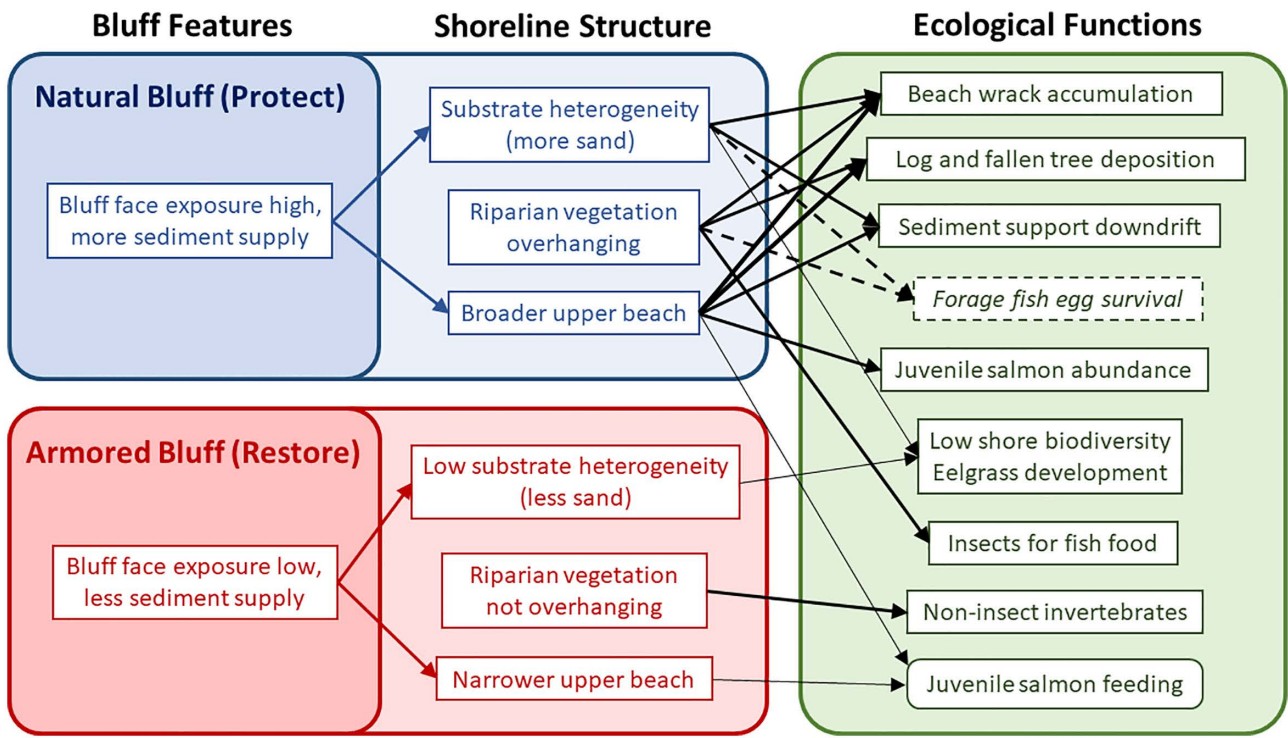

**Fig 7. Conceptual model of bluff features, shoreline structure, and ecological functions.** Showing the relationship between natural (blue box) and armored (red box) feeder bluff features, shoreline structure, and localized beach ecosystem function (green box). Our study showed distinct differences between natural (Protect) and armored (Restore) bluff features and shoreline structure, which influenced the ecological function of the beach. White boxes with solid lines represent metrics that were measured, and arrows represent conceptual relationships between the metrics. Thin arrows suggest weak relationships and thick arrows show statistically significant relationships. There were more juvenile salmon at natural bluffs, however the salmon that were at armored bluffs were more often observed to be feeding offshore, albeit with a low number of total observations for determining direct relationships. Dashed boxes and arrows represent metrics and relationships that were not measured in the present study, but have previously been studied [15].

understanding of the functions that these shoreline habitats can provide, we can better plan for restoration actions that will be sustainable, and enhance coastal resiliency in the wake of global climate change such as sea level rise [5–7].

## Supporting information

**S1 Fig. Boxplots of vegetation, supratidal invertebrates, geomorphology, MLLW biota, and fish metrics with non-significant Treatment differences.** Data points are color-coded by Region, and are minimally staggered to avoid overlap.
(TIF)

**S2 Fig. Boxplots of geomorphology and sediment metrics with nonsignificant Treatment differences.** Data points are color-coded by Region, and are minimally staggered to avoid overlap.
(TIF)

**S3 Fig. Line plots of predictors included in mixed effects models.** Response variables are proportion wrack cover, log line width, count of logs, count of partially buried logs, and log line width. 95% confidence intervals are shaded gray.
(TIF)

**S4 Fig. Average fish composition across treatments from snorkel surveys in spring and summer.**
(TIF)

**S5 Fig. Count of the top 20 vegetation species measured by presence at the twenty study sites.** Colors show native, introduced, and cryptic status.
(TIF)

**S6 Fig. Multivariate NMDS plot of supratidal invertebrates captured in fallout traps.** Symbols are the four treatments, with vectors of taxa correlation. PERMANOVA shows treatment is significant ($p < 0.0001$), with all pairwise comparisons different from each other ($p < 0.05$) except Protect High and Low.
(TIF)

**S7 Fig. Multivariate NMDS plot of MLLW biota at the five regions.** Vectors are taxa and sediment correlation.
(TIF)

**S8 Fig. Bluff/armoring toe to MLLW beach slope (top), and wave height (bottom).** Visualized by geographic region and treatment.
(TIF)

**S1 Table. Site details and location.** Potential benefits of restoration or protection to nearshore sediment supply processes were calculated by [10] using a sum of eight metrics from spatial data that collectively estimate the downdrift benefits of the bluff, based on parameters such as location within the drift cell and proportion of the total sediment supply to the drift cell. Process degradation was the percent of bluffs armored in a drift cell [10]. Treatments with an asterisk were labeled "moderate" by Beach Strategies [10] as they were near low rankings, and were included to expand site selection options across a broader numerical range of potential benefits within a region. Potential sediment supply benefit was higher and process degradation lower at the high ranked sites, and vice versa for the low ranked sites.
(XLSX)

**S2 Table. Metrics included in our scale bar and analyses.** Each metric includes a rationale for investigating feeder bluff and beach function.
(XLSX)

**S3 Table. Post-hoc pairwise tests on Treatment.** Includes Tukey HSD tests of ANOVAs and pairwise comparisons of estimated marginal means from GLMMs. Pairwise comparisons subtract the estimated marginal mean of the second treatment from the estimated marginal mean of the first. Treatment pairs include Protect High (PH), Protect Low (PL), Restore High (RH), and Restore Low (RL). Values include chi-square values, marginal means comparisons, 95% confidence intervals, and p-values.
(XLSX)

**S4 Table. Average lengths of main fish groups during snorkel surveys.** Includes percent of observations at each water column position of surface, middle, and bottom.
(XLSX)

**S5 Table. Percent of main fish groups feeding during snorkel surveys.** Includes where they were feeding in the water column.
(XLSX)

**S6 Table. Descending amount of percent total linear extent of overhanging vegetation across all regions and treatments.**
(XLSX)

## Acknowledgments

Fieldwork help included Evan Lahr, Bob Oxborrow, Arielle Tonus Ellis, and Cormac Toler-Scott. Thanks to Nino and Erika Ramirez, Karen and Richard Person, David Mallory, Christine Garrigan, David Bernhard, Nicolas Duchastel, Christina Kereki, Sheila Thomas, Hartstene Pointe HOA, and Cliffside Beach HOA for facilitating access to beach properties. Aspects of site selection and methods refinement were discussed with our technical advisory group: Tish Conway-Cranos, Jay Krienitz, Andrea MacLennan, Doris Small, George Kaminsky, Hannah Faulkner, and Sydney Fishman.

## Author contributions

**Conceptualization:** Jason Toft, Megan N. Dethier, Andrea S. Ogston.

**Data curation:** Jason Toft, Julia N. Kobelt, Kerry L. Accola, Megan N. Dethier, Andrea S. Ogston, Sarah E. Vollero.

**Formal analysis:** Jason Toft, Julia N. Kobelt, Kerry L. Accola.

**Funding acquisition:** Jason Toft, Megan N. Dethier, Andrea S. Ogston.

**Investigation:** Jason Toft, Julia N. Kobelt, Kerry L. Accola, Megan N. Dethier, Andrea S. Ogston, Sarah E. Vollero.

**Methodology:** Jason Toft, Julia N. Kobelt, Kerry L. Accola, Megan N. Dethier, Andrea S. Ogston, Sarah E. Vollero.

**Project administration:** Jason Toft.

**Supervision:** Jason Toft, Megan N. Dethier, Andrea S. Ogston.

**Writing – original draft:** Jason Toft, Megan N. Dethier, Andrea S. Ogston.

**Writing – review & editing:** Jason Toft, Julia N. Kobelt, Kerry L. Accola, Megan N. Dethier, Andrea S. Ogston, Sarah E. Vollero.

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
