## [Decision Letter · Decision Letter 0]

24 Jul 2025

Dear Dr. Toft,

Thank you for submitting your manuscript to PLOS ONE. After careful consideration, we feel that it has merit but does not fully meet PLOS ONE’s publication criteria as it currently stands. Therefore, we invite you to submit a revised version of the manuscript that addresses the points raised during the review process.

Two reviewers have provided helpful feedback for your revisions. The majority of the reviewers' recommendations focus on improving clarity and providing adequate context and other information for readers to follow you through to your conclusions. One reviewer suggested shortening the article. To clarify, PLOS does not have length restrictions but does encourage presenting information concisely.

We look forward to receiving your revised manuscript.

Kind regards,

Stephanie S. Romanach, Ph.D.

Academic Editor

PLOS ONE

Journal Requirements:

2. We note that Figure 1 in your submission contain [map/satellite] images which may be copyrighted. All PLOS content is published under the Creative Commons Attribution License (CC BY 4.0), which means that the manuscript, images, and Supporting Information files will be freely available online, and any third party is permitted to access, download, copy, distribute, and use these materials in any way, even commercially, with proper attribution. For these reasons, we cannot publish previously copyrighted maps or satellite images created using proprietary data, such as Google software (Google Maps, Street View, and Earth). For more information, see our copyright guidelines: http://journals.plos.org/plosone/s/licenses-and-copyright.

Reviewers' comments:

Reviewer's Responses to Questions

**Comments to the Author**

1. Is the manuscript technically sound, and do the data support the conclusions?

Reviewer #1: Yes

Reviewer #2: Partly

2. Has the statistical analysis been performed appropriately and rigorously?

Reviewer #1: Yes

Reviewer #2: Yes

3. Have the authors made all data underlying the findings in their manuscript fully available?

Reviewer #1: Yes

Reviewer #2: Yes

4. Is the manuscript presented in an intelligible fashion and written in standard English?

Reviewer #1: Yes

Reviewer #2: Yes

Reviewer #1: The paper presents a comprehensive analysis of beach functions provided by cliffs of Puget Sound, WA, USA. The vast amount of data covering a wide range of beach functions was analyzed, which is particularly worth commendation, as it provided grounds for a genuinely in-depth study geared toward possible management strategies regarding sediment availability as one of key factors of sustainable coastal zone management. In this context the Authors determined four types of possible strategies, two of which describe the action priority (high vs. low) and two other the action type (protect – i.e. prevent armoring, and restore, i.e. discard armoring). The integration of multiple datasets and their subsequent joint analysis with consistent metrics can therefore be recommended for similar studies at other shores; the paper definitely deserves publication as source reference for such studies. However, it is focused solely on the Puget Sound, so the presented methodology cannot be readily implemented at other sites due to specific features and constraints of that basin. Consequently, the results gained herein need to be generalized for other sites in the discussion and conclusions sections before the paper can serve as valuable source reference. Thus, I recommend a minor revision of the manuscript. The specific issues to be amended are listed below.

1. The Authors confine their reflections to two basic management options regarding sediment budgets at diverse locations at Puget Sound: leaving coastal bluffs undisturbed to secure sediment supply (protection) and discarding man-made structures to unblock that supply (restoration). However, I found no reflection on why some coastal units were armored – only in L. 46-47 coastal squeeze in UK is mentioned as detrimentally impacting beach functions there. I believe omission of the human factor and the concentration on ecological function in Puget Sound produces somewhat insufficient description of the study area. We do not know what human uses forced armoring projects (large investments?, land development?, harbor construction?, recreation?, etc.). This information is necessary to gain a more comprehensive understanding of the underlying main coastal zone management concepts in the study area.

2. One of such concepts is linked to potential implications of reestablishment of sediment supply by considering maximum permissible (annual) cliff retreat vs. the functions of hinterland. Is this concept included in the preparation of local coastal management plans/strategies?

3. Human interventions did take into account local beach functions by concentrating the structures near higher beach segments ( L. 466-468). Taking into account the geomorphology of Puget Sound (short fetches) I believe this basin is dominated by tide-induced currents and wave action is a secondary source of energy. However, there are numerous cliffs, where the waves are a predominant driver of coastal change. Coastal protection/management is much more challenging at such sites and the armor must be more robust, embracing lower beach portions as well. Moreover, the Puget Sound cliffs are generally composed of coarse(r) grains, offering some resistance against migration/depletion. By contrast, there are multiple cliffs made predominantly of sand, sometimes with insertions of lenses of organic soils, or containing layers of tills, silts or clay that together can form very complicated ‘sandwiches’. Sustainable management of such cliffs and restoration of their beach functions appears to be much more difficult, as mere removal of armoring may often lead to unacceptable damages to hinterland and public protests. The Authors are asked to consider these aspects in their reflections in the discussion and conclusions sections and thus achieve a more general perspective.

4. The Authors seem to be preoccupied with ‘passive’ approach to sustainability by considering two basic types of action – protect/restore. I wonder why they did not consider beach fills as the means of recharging local sediment sources. Again, I suspect it is related to the geomorphology of Puget Sound, its short fetches and the resulting energy fluxes from oblique directions, where aggressive, shore-normal wave action events are rare (cf. L. 201-203). At the open coasts cliff maintenance is often done using beach fills, sometimes supported by low-crested breakwaters and/or groins, aimed at minimum overall seascape interference. In such instances either fully nature-based solutions are implemented, or permanent intervention remains minimal (and acceptable). I would expect acknowledging of the fact that such solutions do exist and can maintain/restore beach functions. This remarks is related to L. 58-60, where the sedimentation cycle is mentioned – please note that at open coasts the sediments are often carried offshore and lost forever. This aspect is also important to highlight to what extent fully nature-based solutions at Puget Sound are feasible (e.g. by sediment recycling with beach fills – if possible at Puget Sound this solution should minimize the costs of nourishments).

5. Following the previous point I found no precise information about the sources of sediment from bluffs – implicitly it becomes available due to marine processes of erosion. Given the morphology of Puget Sound (long and narrow, interconnected basins) I wonder whether wind directions favor the development of surges at the extremities of long embayments forming the Sound (and contribute to erosion events there)? Second, are heavy rains encountered in the area that can trigger local slides developing by generation of pore pressure in cliff massifs? Third, is there a significant role of ice freezing inside the massifs that can contribute to reduction of cliff stability during spring seasons? These pieces of information would be very helpful for foreign readers.

Reviewer #2: The research topic and findings contribute valuable data to the field. However, several revisions are necessary to improve the clarity and effectiveness of the article in communicating the study’s rationale, goals, and results. Please consider addressing the following concerns:

1. Article Length: While the manuscript contains detailed information, its overall length exceeds typical standards and resembles a project report. It is recommended to shorten the article to improve readability and maintain focus.

2. Introduction Coherence: Certain sections of the introduction lack coherence, which affects readability. For example, the relationship between coastal squeeze, coastal erosion, and flooding (Lines 45–47) is not clearly articulated. Please revise this section to ensure a more logical and cohesive presentation of ideas.

3. Study Area Map (Figure 1): The current map does not clearly depict the beach locations. Replacing it with a clearer map, such as one based on Google Earth or a more legible base map with clearly marked sampling sites, would help readers better understand the geographical context of the study.

4. Figure Quality: The resolution of all figures should be improved to ensure better visual clarity and presentation quality.

5. Line 121 – Use of "Sound-wide": The term "Sound-wide" may be unclear to readers unfamiliar with the region. If it refers to Puget Sound or another specific location, please clarify by explicitly stating the geographic name (e.g., "Puget Sound-wide") to improve accessibility for a broader audience.

6. Materials and Methods – Line 118: Please include the site selection methodology and the rationale for selecting the 30 metrics in two clearly separated sections. It is important to introduce the full set of metrics and explain the reasoning behind their selection before discussing the data collection process. Additionally, refer readers to Table 1 for the complete list of metrics at this point, so they are familiar with the variables being analyzed before proceeding further into the methods.

7. Line 122 – Reference to S1 Table and External Report: Referring readers to S1 Table and an external report for key methodological information is not ideal in an article that aims to independently present its methods, background, and results. It is recommended to revise the caption for S1 Table to include an explanation of how the different ranks and value ranges should be interpreted (e.g., which ranges are considered low, moderate, or high).

8. Line 182 – First Use of "MLLW": I believe this is the first instance where "MLLW" appears in the text. Please spell out the full term.

9. Line 206. Did the analysis of ecological functions account for the age of shoreline armoring at the selected sites? Specifically, was the timing of when each site was armored considered? Do you think this factor could influence the assessed value of beach functions, and if so, how?

10. Line 253. In Table 1, please spell out the abbreviations PH, PL, RH, and RL to ensure clarity for readers.

11. Line 552. While the analysis, comparison, and ranking presented in this study emphasize ecological functions and sediment supply across the selected sites, and explore the benefits of restoration approaches, it remains unclear what the existing shoreline armoring is currently protecting, what the key functions of the studied shorelines are, and why they are important. Were these structures originally installed to safeguard properties, infrastructure, or other assets along developed coastlines? Clarifying the current role of the armored bluffs, along with the potential costs and benefits of the recommended actions, such as armor removal or the implementation of green shoreline approaches, would help readers better understand the rationale behind the study’s recommendations.

12. As the authors also noted, the physical, hydrological, and ecological characteristics of the 20 selected sites can vary significantly. How was this variability addressed to reduce uncertainty and potential error in the comparison results? It is recommended that the authors add a dedicated section to discuss the methodological limitations of the study, including potential sources of error and uncertainty in the analysis.

I hope these comments assist the authors in enhancing the presentation of their research outcomes.

Regards,

**Do you want your identity to be public for this peer review?** For information about this choice, including consent withdrawal, please see our Privacy Policy

Reviewer #1: **Yes: ** Grzegorz Różyński, Polish Academy of Sciences, Institute of Hydro-Engineering, Gdańsk, Poland

Reviewer #2: No

---

## [Author Response · Author response to Decision Letter 1]

25 Sep 2025

Please see uploaded "Response to Reviewers"

---

## [Editor Report · Decision Letter 1]

1 Oct 2025

Functions of coastal feeder bluff systems: implications for prioritizing protection and restoration

PONE-D-25-22073R1

Dear Dr. Toft,

We’re pleased to inform you that your manuscript has been judged scientifically suitable for publication and will be formally accepted for publication once it meets all outstanding technical requirements.

Kind regards,

Stephanie S. Romanach, Ph.D.

Academic Editor

PLOS ONE
---

## [Editor Report · Acceptance letter]

PONE-D-25-22073R1

PLOS ONE

Dear Dr. Toft,

I'm pleased to inform you that your manuscript has been deemed suitable for publication in PLOS ONE. Congratulations! Your manuscript is now being handed over to our production team.

Kind regards,

on behalf of

Dr. Stephanie S. Romanach

Academic Editor

PLOS ONE